

# A thermodynamic description for the hygroscopic growth of atmospheric aerosol particles

Dimitri Castarède[1,2] and Erik S. Thomson[1]

[1]Department of Chemistry and Molecular Biology, Atmospheric Science, University of Gothenburg, Gothenburg, Sweden
[2]Previously at, Observatoire Midi-Pyrenees, University of Toulouse (Paul Sabatier, Toulouse III), France

**Correspondence:** E.S. Thomson (erik.thomson@chem.gu.se)

**Abstract.** The phase state of atmospheric particulate is important to atmospheric processes and aerosol radiative forcing remains a large uncertainty in climate predictions. That said, precise atmospheric phase behavior is difficult to quantify and observations have shown that "precondensation" of water below predicted saturation values can occur. We propose a revised approach to understanding the transition from solid soluble particles to liquid droplets, typically described as cloud condensation nucleation – a process that is traditionally captured by Köhler theory, which describes a modified equilibrium saturation vapor pressure due to I. mixing entropy (Raoult's law) and II. droplet geometry (Kelvin effect). Given that observations of precondensation are not predicted by Köhler theory, we devise a more complete model which includes interfacial forces giving rise to predeliquescence, i.e., the formation of a brine layer wetting a salt particle at relative humidities well below the deliquescence point.

*Copyright statement.* TEXT

## 1 Introduction

The role of aerosols in the radiative budget of the planet is a source of large uncertainty in climate modeling and prediction (Stocker et al., 2013). One significant source of uncertainty comes from inadequate understanding of aerosol phase state in the atmosphere (Davis et al., 2015b). The phase behavior of atmospheric particles depends on both the environmental conditions (pressure, temperature, humidity, etc.) and the particle properties (Fig. 1). The phase state influences surface as well as bulk phase chemistry, cloud forming potential, particle deposition and other aspects of the global water and bio-/geo-chemical cycles. Thus fundamental knowledge of aerosol particle phase behavior is a key aspect to understanding and modeling atmospheric processes. For example, mixed phase clouds in the Arctic demonstrate a surprising persistence that is not predicted by the current understanding of ice-water saturation vapor pressure gradients and the resulting competition for $H_2O$ (Wegener-Bergeron-Findeisen process) (Korolev, 2007; Martin et al., 2011; Morrison et al., 2012).

For soluble atmospheric particulate, Köhler theory is generally used to quantify and parameterize atmospheric phase state (Köhler, 1936). Köhler theory describes the equilibrium size of solution droplets in the atmosphere as determined by the satu-



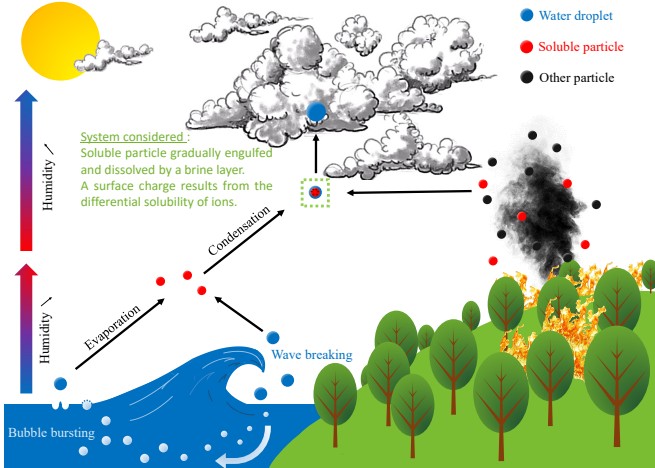

**Figure 1.** Soluble particles are ubiquitous in the atmosphere, with primary sources including biomass burning and the sea surface illustrated here. Solid salt particles, which can be directly emitted or form when solution droplets evaporate and effloresce, provide hygroscopic surfaces for cloud condensation nucleation.

ration vapor pressure, which depends upon component mixing (Raoult's Law) (Pruppacher and Klett, 1997) and the curvature of the interface (Kelvin or Gibbs-Thomson effect) (Thomson, 1871; Orr et al., 1958). For a fully soluble particle, implicit in the theory is a sudden transition from a dry (solid) particle to a saturated droplet. The relative humidity at the transition point is referred to as the deliquescence relative humidity (DRH). The Köhler model is useful because it provides a simple physical and mathematical description of nucleated condensation and can be modified to include compounds of limited solubility (Bilde and Svenningsson, 2004). As such it has remained the tool of choice in atmospheric models (Mirabel et al., 2000; Nenes and Seinfeld, 2003), although many measurements suggest that a precondensation of water even on pure soluble surfaces may occur below the DRH (Hämeri et al., 2001; Biskos et al., 2006; Zeng et al., 2013; Davis et al., 2015a; Cheng et al., 2015; Montgomery et al., 2015; Hsiao et al., 2016). For pure compounds, such observations are not predicted and thus hint that Köhler theory can be refined.

Here we suggest a theoretical refinement by considering the stability of a salt particle that is gradually engulfed and dissolved by a brine layer (Fig. 2). The model includes the understood bulk phase equilibria established by Köhler theory but invokes a transition region where meta-stable liquid layers exist on particle surfaces. The general shortcoming of Köhler theory for single component systems is the activated transition from dry particle to liquid droplet and the exclusion of an interfacial system that reduces the global free energy. The theory we propose herein evolves from previous explorations of wetting of soluble surfaces (Russell and Ming, 2002), where previous work has used bulk thermodynamics and ascribed disjoining pressures (also modeled using bulk properties or fitting parameters) to explore the stability of thin films on solvating particles (Shchekin and Rusanov, 2008; Shchekin et al., 2008; McGraw and Lewis, 2009; Shchekin et al., 2013). Other studies have used similar theoretical models to treat particle interfaces in the presence of substrate surfaces (Bruzewicz et al., 2011), or for terrestrial systems





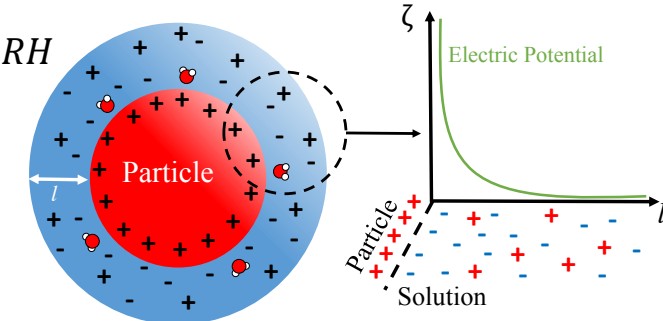

**Figure 2.** Schematic of the system considered herein – an idealized solvating atmospheric particle. The use of an idealized spherical geometry is justified by previous findings that NaCl crystal corner and step sites dissolve preferentially to surface sites, in a manner that quickly roughens and rounds faceted crystals (Chen et al., 2014).

like brine crusts where geometric effects can be excluded (Hansen-Goos et al., 2014). In this treatment we include terms to explicitly account for the intermolecular origins of the interfacial forces in a manner akin to surface melting (Dash et al., 2006).

Thus we extend previous work to combine and define, entropic, geometric and explicit intermolecular interactions that result in modified equilibrium vapor pressures above particle surfaces. We do so in a manner wherein achievable analytic solutions help to illuminate the underlying physicochemical processes that control the system, and may be used to benefit modeling at a range of scales. The implications of this work are far-reaching given that within the current state of understanding the existence of atmospheric aqueous surfaces is limited to deliquesced materials and/or coated insoluble particles. The model we propose herein suggests that aqueous surface layers may exist in some range of relative humidity (RH) below the DRH and thus resemble liquid droplet behavior for some processes and detection techniques.

## 2    Refinement of Köhler Theory

Köhler theory considers and balances the effects on droplet stability due to the interfacial curvature and soluble components (Farmer et al., 2015; Ruehl et al., 2016; Lohmann et al., 2016). Thus the equilibrium vapor pressure $p_r$ at temperature $T$ above a brine droplet of radius $r$ is described by its departure from the analogous vapor pressure at the flat, pure water interface $p_{b\infty}$ at the same temperature,

$$\frac{p_r}{p_{b\infty}} = \frac{p_{sol}}{p_{b\infty}} \cdot \frac{p_r}{p_{sol}}, \tag{1}$$

where $p_{sol}$ represents the equilibrium vapor pressure over the flat brine solution surface, also at temperature $T$. In detail the geometric term in (1) is given by the Kelvin equation,

$$\frac{p_r}{p_{sol}} = \exp\left(\frac{2\sigma v_{sol}}{R_g T r}\right), \tag{2}$$





where the surface free energy of the liquid/gas interface $\sigma \equiv \sigma(\rho_{sol})$ is a function of the bulk composition of the solution droplet as expressed by the solution density $\rho_{sol}$, $R_g$ is the ideal gas constant, and $v_{sol}$ is the molar volume of the solution. Likewise the modified equilibrium due to the addition of soluble components is given by the water activity,

$$\frac{p_{sol}}{p_{b\infty}} = a_w \qquad (3)$$

where a theoretical model (e.g., Van't Hoff, E-AIM) or empirical parameterization can be chosen to best match the system of interest.

Thus from the combination of these effects the DRH is implicitly given when $r = \mathrm{R_{DRH}}$ at the radius of the deliquesced droplet,

$$\frac{\mathrm{DRH}}{100} = a_w \exp\left(\frac{2\sigma v_{sol}}{R_g T \mathrm{R_{DRH}}}\right). \qquad (4)$$

This traditional Köhler formulation (4) for a fully soluble particle predicts the deliquescence point and, when $r$ is allowed to evolve above $\mathrm{R_{DRH}}$, the equilibrium relative humidity above the evolving brine droplet. However, in this formulation the sudden transition from a solid to liquid particle remains implicit.

Although the transition from solid to liquid solution may proceed quickly, it is important to consider whether or not there exists an intermediate state(s) of importance. Thus we consider whether a particle may be wetted by a thin film and what, if

any, stability conditions may (de-)stabilize such a system. Such thin films are a common phenomenon and are generally treated within the rubric of adsorption or wetting (Schick, 1990), fields which incorporate many important applications from biology to surfactant physics (French et al., 2010).

Reformulating atmospheric particle dissolution as an interfacial problem leads to the system postulated in Fig. 2, where a charged soluble particle is engulfed and dissolved by a brine layer. However, in order for the interfacial system to endure it

must yield some energetic benefit that can be captured by minimizing the global free energy of the system. Thus in addition to the bulk free energies tacit in Köhler theory (2) – (3) an interfacial contribution $\zeta(l)$ must be included. For thin films Derjaguin-Landau-Verwey-Overbeek (DLVO) theory, which assesses the balance between short and long range intermolecular interactions, has been used to great success (Derjaguin and Landau, 1993; Verwey and Overbeek, 1948; Luo, 2007).

In the case of an ionic electrolyte a surface charge results from the differential solubility of ions (Kobayashi et al., 2014),

while the solvated ions affect both the mixing entropy and the electric potential of the film. The ions in the brine layer are organized to offset the surface charge in a manner described by linearized Poisson-Boltzmann theory, where the characteristic falloff of the electric field is a *Debye Length* $\kappa^{-1} = \left((\epsilon\epsilon_o k_b T)/(e^2 N_A \rho_{sol})\right)^{1/2}$, where $\epsilon_0$ is the vacuum permittivity, $\epsilon$ is the relative permittivity of the brine, $k_b$ is the Boltzmann constant, $T$ is the absolute temperature, $e$ is the elementary charge, and $N_A$ is Avogadro's constant. Thus the contribution to the free energy is,

$$F_{elec}(d) = \frac{2q_s^2}{\kappa\epsilon\epsilon_0} e^{-\kappa l}, \qquad (5)$$

where $q_s$ is the surface charge and $l$ is the thickness of the liquid layer. The colligative effect of the ions remains unchanged as in (3).



The long range dipole fluctuations in the system are volume-volume interactions and can be depicted to first-order as the non-retarded dispersion or van der Waals forces. Assuming that retardation is not important the van der Waals contribution to the free energy can be expressed as,

$$F_{disp}(d) = -\frac{A_h}{12\pi l^2},\tag{6}$$

where the Hamaker constant $A_h$ is determined for a given layered system (French, 2000). This formulation assumes a planar geometry, which given the relative scales of the layering versus the system size is taken as accurate to first order (French, 2000). For small nano-particles other considerations including the entire particle volume may become important as we discuss later.

     The combined effects of the short and long range interactions captured by $F_{elec}(l)$ and $F_{disp}(l)$ yield a total interaction potential

$$\zeta(l) = F_{disp}(l) + F_{elec}(l),\tag{7}$$

whose derivative with respect to $l$, $\zeta'(l)$ represents the interfacial contribution to the free energy, and whose sign and strength will depend upon the material properties and geometry of the specific system.

     To re-express Köhler theory including the effect of these intermolecular interactions we define an equilibrium vapor pressure over the curved brine surface $p_\zeta$ that includes the interfacial term, and thus in analogy to (1),

$$\frac{p_\zeta}{p_{b\infty}} = \frac{p_{sol}}{p_{b\infty}} \cdot \frac{p_r}{p_{sol}} \cdot \frac{p_\zeta}{p_r}.\tag{8}$$

     Assuming that the thermodynamic equilibrium condition at the solution – vapor interface is set by the free exchange of water molecules $\frac{\partial G}{\partial N_v} = \frac{\partial G}{\partial N_l}$ and that near deliquescence temperature is constant, an expression for $p_\zeta$ can be calculated by considering the difference in chemical potentials between the solvated and layered states,

$$p_\zeta = p_r \exp\left(\frac{v_{sol}\zeta'(l)}{R_g T}\right),\tag{9}$$

where the details of the derivation as presented in Hansen-Goos et al. (hereafter, HG14) (Hansen-Goos et al., 2014) are included in the supplementary material for completeness. Thus (8) can be rewritten as,

$$\frac{p_\zeta}{p_{b\infty}} = \frac{RH}{100} = a_w \exp\left(\frac{2\sigma v_{sol}}{R_g T r'}\right)\exp\left(\frac{v_{sol}\zeta'(l)}{R_g T}\right),\tag{10}$$

$$\text{where,} \quad r' < \text{R}_{\text{DRH}} \rightarrow r' = \text{R}_s + l - \frac{l}{\beta},\tag{11}$$

$$r' \geq \text{R}_{\text{DRH}} \rightarrow r' = r,\tag{12}$$

Thus at $r' < \text{R}_{\text{DRH}}$ the system size evolves like $\text{R}_s + l - \frac{l}{\beta}$, which captures the change in radius due to condensation and dissolution, where a linear solvation is assumed. The exact solution considering the volume/volume equivalence of dissolution requires numerically solving a $3^{\text{rd}}$ degree polynomial and yields a negligible correction factor. The growth factor at DRH is





$\beta = \mathrm{R_{DRH}}/\mathrm{R_s}$, where the dry particle radius is $\mathrm{R_s}$. At $r' \geq \mathrm{R_{DRH}}$ the entire particle is dissolved and thus $r$ is the solution droplet's radius and the theory re-converges to classical Köhler theory as $\zeta'(l)$ vanishes.

Equation (10) is a general result describing the equilibrium vapor pressure over a dissolving salt particle, from the dry particle state to the totally dissolved state. Although herein we treat an idealized monovalent electrolyte system using modified DLVO

theory to constrain the functional behavior of $\zeta'(l)$, natural systems may require more complex treatments that would likely yield a host of interesting behavior, and simultaneously strain the ability to achieve analytical and/or computational solutions.

## 3   Applying refined Köhler theory

It is instructive to use the refined Köhler formulation (10) to model a NaCl particle, as might represent an idealized marine aerosol. Although considerable information concerning bulk salt solutions is available it is difficult to assess the applicability of

these values to thin brines. For example, the assumptions that the interfacial brine layer is a saturated solution whose thickness is controlled by electrostatic interactions may not be self-consistent. The ion availability within a saturated NaCl brine will allow efficient charge screening and thus a very short Debye Length should result. That said the uncertain theoretical parameters, $q_s$, $A_h, \kappa^{-1}$, and brine concentration $C$ can also be used as fitting parameters in order to illuminate the range of possible physical behavior.

Here for an idealized case we choose to apply (10) to a sodium chloride particle of a representative atmospheric diameter $\simeq 0.8\,\mu$m in the accumulation mode (Lewis and Schwartz, 2004; Lohmann et al., 2016). The growth factor is assumed such that the particle will lead to a solvated brine droplet of radius $1.36\,\mu$m, and for consistency with previous work we choose a temperature of $20°$ C and a saturated concentration of $[\mathrm{NaCl}]_{sat} = 5.4\,\mathrm{mol\,L^{-1}}$ (Haynes, 2012), which is also used to calculate the Debye length. The value for surface charge $q_s = 0.12\,\mathrm{C\,m^{-2}}$ is taken from Kobayashi et al. (2014), while the HG14 value

$A_h = -1.5 \times 10^{-20}$ is used, and the expression for the water activity of a NaCl solution is taken from Tang et al. (1997). The result is illustrated in Fig. 3, where both the classical Köhler behavior and the refined model of pre-deliquescent hygroscopic growth are captured. With the refined interfacial model the DRH is captured ($\approx 75\%$ RH) as reported in many previous studies (Tang and Munkelwitz, 1994; Metzger et al., 2012; Laskina et al., 2015), but near the deliquescence transition (Fig. 3, zoom) a wetted interface is predicted below DRH when considering the balance of the intermolecular interactions.

The result suggests that observations of pre-condensation in the existing literature (Hämeri et al., 2001; Biskos et al., 2006; Cheng et al., 2015) may also be explained by a metastable interfacial equilibrium. In Fig. 4 the theoretical model is compared to measurement data from Hämeri et al. (2001) and Biskos et al. (2006). Each of the solid lines in Fig. 4 correspond to non-linear least-squares solutions to fit the data, where the identified fitting parameters are presented in Table 1.

For physical consistency the best-fit solutions have been calculated excluding data points that preclude a full monolayer of

water ($l \leq 0.3$ nm) and excluding the data points that represent the fully solvated particles (Growth Factor $\geq 1.2$ in Fig. 4). In all cases the data is well represented within a narrow window of the fitting parameters and the best-fit solutions also serve to demonstrate the impact of the parameters on the shapes of the curves. Furthermore, the best fit solutions agree well with values



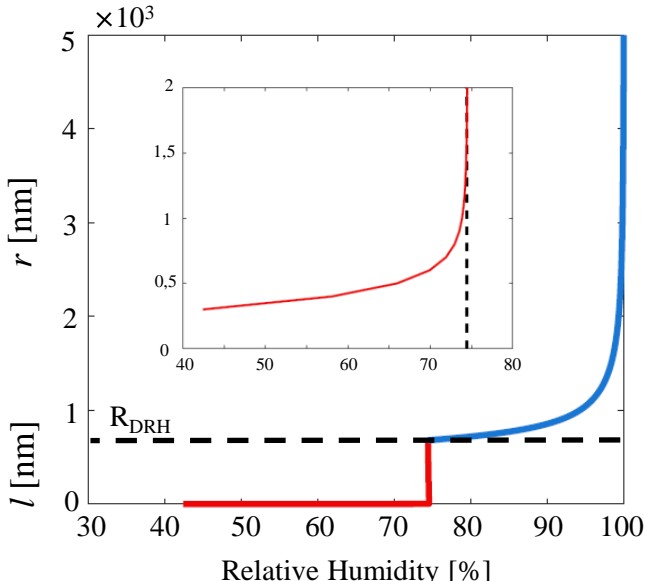

**Figure 3.** Model of the growth of the solvating surface of a 0.8 $\mu$m accumulation mode NaCl particle as outlined in the text. Below deliquescence ($\lesssim 75\%$ RH and in zoom) the metastable brine layer behavior is governed as in Eq. (10) – red curve. Above deliquescence the layer thickness $l$ becomes the radius $r$ of the growing solution droplet as is captured by classical Köhler theory (Eqs. (1)–(4) and Eq. (10) for $r' = r$, where the interfacial term has vanished; blue curve).

extracted from the literature, listed in the first row of Table 1 and used previously to construct Fig. 3, where their sources are referenced.

For small particles ($R_s \lesssim 5$ nm) the Kelvin term is strong enough to compete with the intermolecular forces and thus the system retains an activation barrier until DRH or above (McGraw and Lewis, 2009), as shown in Figs. 4–5. A small dry particle would first be subjected to reversible uptake of water due to intermolecular attraction until it suddenly dissolves into a brine droplet when the deliquescence activation barrier is overcome. However, at short length scales the veracity of the bulk approximations and several other simplifying assumptions of this model must be questioned. For example, the high brine layer concentrations predicted for small particles may be indicative of the model limitations. Although the model represents the data remarkably well even where it might be expected to fail, in those cases it might be better to approach the activation problem in terms of adsorption theory (Langmuir, 1918) or using molecular dynamics simulations (Lovrić et al., 2016).

The refinement of Köhler theory we have proposed, yields a smooth meta-stable transition from solid to aqueous phase atmospheric particles. It captures observed behavior for specific compounds, yet remains general such that its application to more complex systems may yield deeper understandings of aerosol phase state and particle behavior.





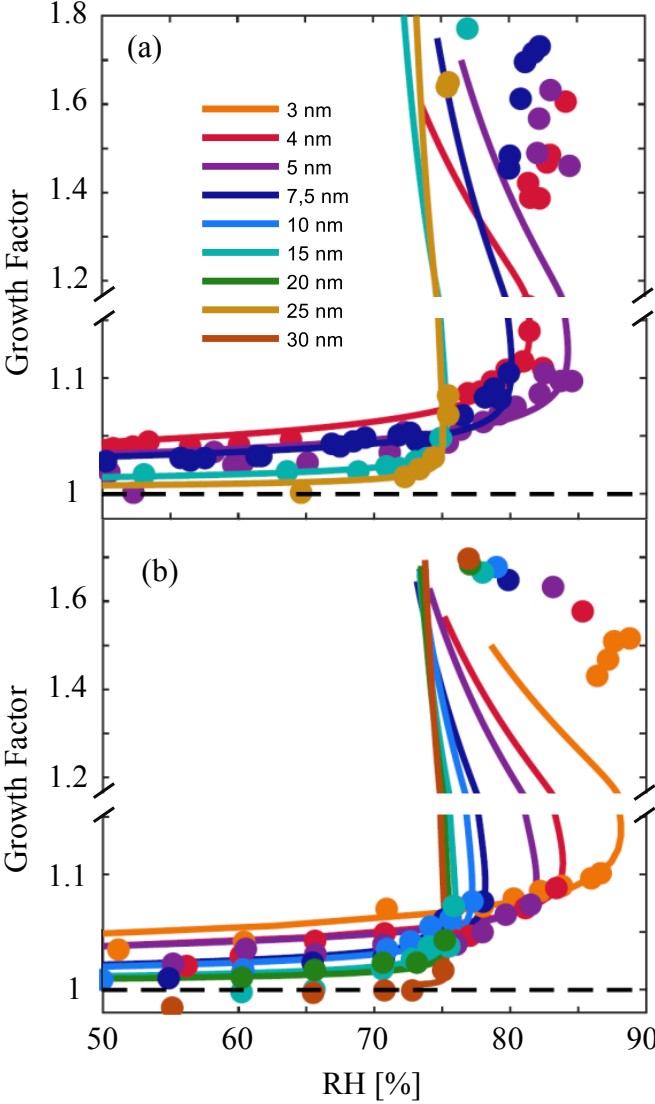

**Figure 4.** Measurement data points of NaCl particle hygroscopic growth under incrementally increasing RH. Colors correspond to different initial dry particle radii indicated in the legend and the growth factor is the ratio of the measured radius to the dry radius. (a) Lines represent theoretical fits to data presented by Hämeri et al. (2001). (b) Theoretical fits to data presented by Biskos et al. (2006). In all cases fitting parameters are presented in Table 1. Given the experimentally prescribed increasing humidities, particles that grow beyond the activation barrier set by the strong Kelvin effect (cf. Fig. 5) are observed at RH exceeding the equilibrium value.



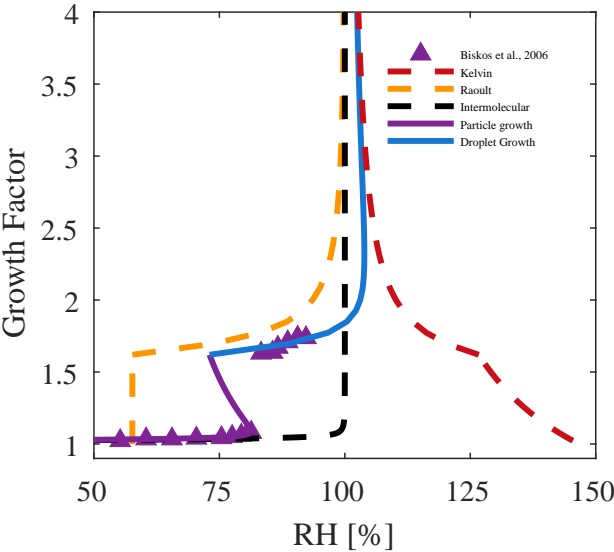

**Figure 5.** The evolution of the individual terms in Eq. (10) are shown for a solvating 5 nm NaCl particle. For such small particles the Kelvin term dominates yielding an activation barrier as illustrated by the inflection in the layer growth (magenta) curve. One also observes that the Raoult term (yellow curve) is onset only after the solid particle dissolves ($R_{DRH}$) and that the intermolecular interactions are very short range (black curve).

## 4 Conclusions

Real particles in the atmosphere tend to be more complicated than idealized theories of solvation can easily capture. Atmospheric particulates that assist nucleation are often internally mixed and include varying quantities of soluble/insoluble, organic/inorganic materials etc (Zardini et al., 2008; Seinfeld et al., 2016). Furthermore, theoretical adaptations of Köhler the-
5 ory have been used to capture particle mixing state; for example "modified-Köhler theory" (Bilde and Svenningsson, 2004) and "$\kappa$-Köhler theory" (Petters and Kreidenweis, 2007). However, these remain limited to predicting critical supersaturations and droplet evolution. Our contribution is general in that it predicts the complete evolution from the dry particle through a meta-stable equilibrium characterized by a growing thin film. Such films are not only consistent with droplet deliquescence but also with previous observations of water absorption and ionic mobility at RH far below the DRH ($35\% \leq RH \leq DRH$)
(Ewing, 2005; Verdaguer et al., 2005; Wise et al., 2008). The implication is that given the correct intermolecular force balance, the surface of any soluble-material-containing atmospheric particle may "pre-deliquesce" and thereby contribute to an as yet unquantified aqueous reservoir.

We note that given the approximations inherent in the model, the fitted concentrations of the brine layers increase with decreasing particle size. These predicted equilibrium concentrations may seem physically unrealistic for very small particles,
which could be due to the geometric limitations of our model – the dispersion forces are derived assuming interactions between





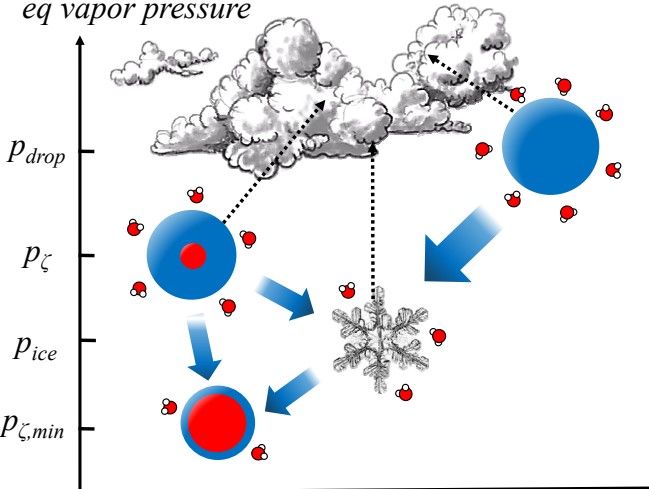

**Figure 6.** Schematic of the equilibrium vapor pressures over droplets, pre-deliquesced particles, and ice particles where gradients lead to Wegener-Bergeron-Findeisen processing. The range of equilibrium vapor pressures above pre-deliquesced thin films straddle the equilibrium vapor pressure over ice $p_{ice}$, resulting in a more stable co-existance between ice and pre-deliquesced particles, relative to liquid droplets.

flat, parallel interfaces. But it is also possible that the model is compensating for other physical effects not taken into account in this study (such as surface depletion), effects that could lower the chemical potential of the liquid phase for very small sizes. The model does illustrate that due to the enduring activation barrier for small particle dissolution, those particles which manage to exceed the barrier will immediately be in a domain of strong supersaturation relative to their equilibrium. Thus, there may

be implications for non-equilibrium particle growth in the atmosphere.

Given the importance of aerosol phase state there may also be significant implications to even a limited RH range where stable aqueous interfacial films exist. Most obvious is the significance for contributions to aqueous phase chemistry (Martin, 2000). However, there are also potential cloud and climate scale impacts that deserve some investigation. The radiative absorption cross-section of pre-deliquescing particles may significantly change their optical properties as has been shown for aerosol

– particle mixtures (Ackerman and Toon, 1981), and especially for particles that include soot (Jacobson, 2000). There are also implications for understanding mixed phase cloud stability given that the equilibrium vapor pressure above a pre-deliquescence layer can be much lower than the analogous vapor pressure above a liquid droplet (Fig. 6).

Mixed phase clouds are inherently unstable given that air saturated with respect to water is supersaturated with respect to ice, and in fact most precipitation globally originates from mixed phase processes (Mülmenstädt et al., 2015). However,

particularly in the Arctic and sub-Arctic regions the unexplained persistence of mixed phase clouds has consequential climate impacts (Morrison et al., 2012). Simply put the equilibrium vapor gradient innate between supercooled liquid droplets and ice crystals is greatly diminished if droplets are replaced by pre-deliquesced particles (Fig. 6), while to some observational techniques the two morphologies may be indistinguishable. A potential result is slower growing ice crystals and thus a longer lifetime for mixed phase clouds. Although a complete understanding of mixed phase must also involve particle dynamics





(Sullivan et al., 2016), pre-deliquescing particles may play a contributing role and recent studies suggest that there exist more sources for dry soluble particles than previously thought (Davis et al., 2015b).

This study has introduced a refinement of Köhler theory that tracks a soluble atmospheric particle from its dry state to the solvated droplet equilibrium. The model presumes that from molecular scale adsorption to the growth of thin liquid films, the
5  interface can be stabilized by the intermolecular interactions in a system. Although, the details of real atmospheric systems would be subject to a strict bookkeeping, even the highly simplified model proposed here captures many important parameters, like equilibrium vapor pressure and liquid layer thickness, that could contribute to better parameterizations for aerosol-cloud interaction modeling efforts.

*Author contributions.* The authors have contributed equally to this work.

10  *Competing interests.* The authors declare that they have no conflict of interest.

*Acknowledgements.* This research was supported by the Swedish Research Council VR, the Swedish Research Council FORMAS, the Nordic Top-Level Research Initiative CRAICC, and the European Commission for ERASMUS mobility. Hendrik Hansen-Goos, Markus Petters, Sarah Petters, Fabian Mahrt, and Robert McGraw are thanked for helpful discussions. Special thanks to Merete Bilde whose contributions significantly helped to improve the manuscript.





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



**Table 1.** Fitting parameters yielding curves in Fig. 4 where superscripted letters correspond to figure panel.

| $R_s$ | $q_s$ [C m$^{-2}$] | $A_h$ [J] | $C$ [moles m$^{-3}$] |
|---|---|---|---|
| standard values | 0.12 (Kobayashi et al., 2014) | $-1.5 \times 10^{-20}$ (HG14) | 5400 (Haynes, 2012) |
| $3nm^{(b)}$ | 0.3 | $-7.5 \times 10^{-20}$ | 8800 |
| $4nm^{(a)}$ | 0.3 | $-9 \times 10^{-20}$ | 8270 |
| $4nm^{(b)}$ | 0.3 | $-4 \times 10^{-20}$ | 8000 |
| $5nm^{(a)}$ | 0.1 | $-9 \times 10^{-20}$ | 7050 |
| $5nm^{(b)}$ | 0.4 | $-2.5 \times 10^{-20}$ | 7620 |
| $7.5nm^{(a)}$ | 0.6 | $-9 \times 10^{-20}$ | 6700 |
| $7.5nm^{(b)}$ | 0.002 | $-3 \times 10^{-20}$ | 7020 |
| $10nm^{(b)}$ | 0.01 | $-15 \times 10^{-20}$ | 6500 |
| $15nm^{(a)}$ | $2.7 \times 10^{-6}$ | $-5 \times 10^{-20}$ | 6300 |
| $15nm^{(b)}$ | 0.02 | $-4 \times 10^{-20}$ | 6300 |
| $20nm^{(b)}$ | 0.3 | $-5.5 \times 10^{-20}$ | 6000 |
| $25nm^{(a)}$ | 0.1 | $-5 \times 10^{-20}$ | 5850 |
| $30nm^{(b)}$ | 0 | $-0.2 \times 10^{-20}$ | 5800 |