# Peer review of "A thermodynamic description for the hygroscopic growth of atmospheric aerosol particles"

_Atmospheric Chemistry and Physics, 2018_

## Referee Comment (RC1) · Anonymous Referee #2 · 19 Jul 2018

Castarede and Thomson present a theoretical approach to modeling pre-deliquescence condensation of water onto soluble salt particles and particle growth using a "refined Kohler theory". There are implications here for understanding and modeling mixed phase clouds and aqueous chemistry. The authors' theoretical treatment is quite elegant and the manuscript is well written. I am in favor of publication but recommend the authors address the comments below.

Specific comments:

- As the authors note, several other works have been published that also present a theoretical treatment of pre-deliquescence (notably, Bruzewicz et al. 2011). I encourage the authors to more clearly state where their approach differs from others, and the advantages of using the refined Kohler model presented here over that of Bruzewicz et al.

- The inflection in some of the curves in Figure 4 warrants more explicit discussion because this behavior is not physically realistic and represents a limitation of the model. For example, in Fig. 4A, for the purple line, there are two GF values that correspond to 80% RH (at GF~ 1.05 and GF~ 1.3, the inflection point being ~85% RH). I was ultimately able to piece together where the inflection arises, but I encourage the authors to more explicitly discuss the trends shown in Figure 4 and their origins. Specifically, for example, I was looking for a sentence or paragraph in the discussion that explicitly stated "The inflection seen in Fig. 4 for small particles is due to...".

- As is well-known, atmospheric aerosols are not single component and are mixed with organic compounds. The authors mention that "natural systems may require more complex treatments..." (p. 6, line 5) and also that the refined Kohler theory "remains general such that its application to more complex systems may yield deeper understandings of aerosol phase state and particle behavior" (p. 7, line 12), but there is not discussion of steps that may need to be taken to apply the refined Kohler theory to mixed organic/inorganic systems. I encourage more discussion here, as it would facilitate the application of their refinement.

- Have the authors attempted to apply their refined Kohler theory to a system other than NaCl?

- Much of the discussion and implications of the research are mentioned for the first time in the "Conclusions" section. This discussion would be better served in its own section. For example, the discussion surrounding Figure 6, which is currently called out for the first time in the conclusions, would be appropriate in a new section for "Discussion" or "Implications". This would enhance readability.

Technical comments - The y-axis values in the inset to Figure 3 have commas rather

than periods for the decimal point.

- I believe that on p. 1, line 14, the reference Davis et al. 2015b should be labeled as "a" (and Davis et al. 2015a should then be labeled "b").

- On p. 4, line 5, Van't Hoff and E-AIM models are mentioned without references. References should be provided.

---

## Referee Comment (RC2) · Anonymous Referee #1 · 23 Jul 2018

In this study, the formation of a brine layer was proposed to extend the present Koehler theory aiming at predicting the pre-deliquescence water uptake of aerosol particles. The new equation was then used to fit experimental data of NaCl in literature. Using the fitted parameters, the new equation can well reproduce the pre-condensation below the deliquescence point. This study revealed a long-time neglected problem and provided one explanation for the pre-deliquescence water uptake. I believe that this study will bring new insights and trigger more discussions in the community. Thus I would recommend its publication in ACP if my following questions can be adequately addressed.

[Figure]

Comments:

(1) Precondensation threshold

In the fit, the authors "excluded the data that preclude a full monolayer of water (l _ 0.3 nm)". Are you suggesting the monolayer sets a threshold for precondenation, below which the precondensation doesn't occur? Or this is only to optimize the fitting parameters? I'd like to see more clear a structure of precondensation. Thus I would suggest to optimize e.g., extending the scale of Fig. 4 to lower RH (RH<50), and consider using log-scale of (gf-1) in the y axis. Figure 5 is a nice illustration but the visibility of the precondensation (core of this study) is not good.

(2) Non-prompt vs prompt

There has been discussions about the differences between the two experimental dataset used here (Haemeri et al and Biskos et al). Biskos et al. suggested a prompt deliquescence while non-prompt deliquescence was reported in Haemeri et al. That is Biskos found the gf of pre-deliquescenced well below 1.2 while Haemeri showed gf up to ~1.4-1.6 before deliquescence. According to the proposed theory, is such high gf possible from a precondensation?

(3) Prediction

I am thinking how the proposed method can be used to predict the precondensation. Now it seems that you have three adjustable parameters to fit the data, so how many experimental data do you need for a reliable fitting. As shown in Table 1, each size has its own fitting parameters. I am wondering if it is possible to have a universal parameter set that is applicable to all sizes. If so, the applicability of this method will be largely increased.

Technical suggestions:

In Fig. 4, the lines represent the equilibrium RH for each gf based on modified Koehler equation. Since the experimental data shows the equilibrium gf rather than the equilibrium RH, I would suggest plotting the same parameters (equilibrium gf) to avoid confusion.

---

## Author Comment (AC1) · 26 Sep 2018

**Response to referee reports for, "A thermodynamic description for the hygroscopic growth of atmospheric aerosol particles" by Casterède and Thomson**

We would like to thank the reviewers for their constructive and useful comments, which have illuminated several areas for improvement and clarification within our manuscript. We try to answer each reviewer question and utilize the suggestions in order to improve the manuscript, and have prepared a revised manuscript accordingly Below we explicitly address and/or point to changes in the text that address each of the items raised in the comments. The responses are presented in the context of the Reviewer comments (*italicized*), where our responses are preceded by bullet points and changes to the text are highlighted in blue.

To best illustrate the extent of the changes to the text the output of a latex difference file is also attached.

**Anonymous Referee #1**

*Comment (1): Precondensation threshold*

*In the fit, the authors excluded the data that preclude a full monolayer of water ($l \leq 0.3nm$). Are you suggesting the monolayer sets a threshold for precondenation, below which the precondensation doesn't occur? Or this is only to optimize the fitting parameters? Id like to see more clear a structure of precondensation. Thus I would suggest to optimize e.g., extending the scale of Fig. 4 to lower RH (RH< 50), and consider using log-scale of (gf-1) in the y axis. Figure 5 is a nice illustration but the visibility of the precondensation (core of this study) is not good.*

• We do not suggest that a water monolayer is a threshold for adsorption, nor do we assume this condition to optimize fitting parameters. Certainly adsorption begins at the molecular scale and we cite literature in this regard ((Ewing, 2005)). However, implicit within the bulk theoretical formulation we propose is that the electrolyte dissolves into a bulk solution of uniform thickness. Although the theory might be used to estimate sub-monolayer coverages, a limit of our model is that sub-monolayer adsorption is not rigorously resolved.

We have changed the sentence in question to, "For physical consistency with the theoretical framework the best-fit solutions...". Furthermore, we add a reference (Peters and Ewing, 1997) to emphasize the point above.

As justified above the x-axis scales have been chosen to deemphasize the sub-monolayer region. However, the suggestion of adopting a log-scale (GF-1) is appreciated and the figure has been changed accordingly.

*Comment (2): Non-prompt vs prompt*

*There has been discussions about the differences between the two experimental dataset used here (Haemeri et al and Biskos et al). Biskos et al. suggested a prompt deliquescence while non-prompt deliquescence was reported in Haemeri et al. That is Biskos found the gf of pre-deliquescenced well below 1.2 while Haemeri showed gf up to 1.4-1.6 before deliquescence. According to the proposed theory, is such high gf possible from a precondensation?*

• In fact, Biskos et al. (2006) assert that the 'non-prompt' deliquescence observed by Hämeri et al. (2001) originates from an instrumental artifact or error. Moreover it is impossible with the

information given to draw conclusions from the experimental data given the associated uncertainties. However, at high GF (1.4-1.6) the stabilizing effect of the intermolecular interactions becomes very small and we would expect the equilibrium to be a deliquesced state.

*Comment (3): Prediction*

*I am thinking how the proposed method can be used to predict the precondensation. Now it seems that you have three adjustable parameters to fit the data, so how many experimental data do you need for a reliable fitting. As shown in Table 1, each size has its own fitting parameters. I am wondering if it is possible to have a universal parameter set that is applicable to all sizes. If so, the applicability of this method will be largely increased.*

• We have presented the theory in its general form in order to demonstrate both its robustness and flexibility. Given that we hope others will be interested in applying the theory to a variety of systems, it seems most advantageous to maintain the generality. That said, for any particular system absolute parameters can be imposed or chosen to reduce the number of fitting parameters. How well this works depends upon how well those parameters can be independently ascertained and/or constrained. Furthermore, the well defined parameter set may vary between systems. For example, we can do such an analysis for the NaCl system with which we have made comparisons. We deem that discussion outside the scope of the main manuscript, but agree with the referee that it may be of interest to parties interested in utilizing the theoretical framework. Therefore we include a supplementary discussion of the NaCl fitting parameters at the conclusion of this document.

*Comment (4): Technical suggestions*

*In Fig. 4, the lines represent the equilibrium RH for each gf based on modified Koehler equation. Since the experimental data shows the equilibrium gf rather than the equilibrium RH, I would suggest plotting the same parameters (equilibrium gf) to avoid confusion.*

• For clarity, the theoretical lines represent the equilibrium between a given system size (layer thickness/particle size) and saturation condition. Mathematically it is most straightforward to calculate both the RH and GF from specified layer or particle sizes, but the means of calculation does not alter the equilibrium. The plots conform with how the experimental data was originally presented.

**Anonymous Referee #2**

*As the authors note, several other works have been published that also present a theoretical treatment of pre-deliquescence (notably, Bruzewicz et al. 2011). I encourage the authors to more clearly state where their approach differs from others, and the advantages of using the refined Kohler model presented here over that of Bruzewicz et al.*

• We agree with the referee that our work follows and extends previous work from other researchers. Notably Russell and Ming (2002) treat a system as we describe using a simple wetting argument to investigate the total free energy of a solid/liquid, liquid/vapor system versus a single interface solid/vapor system. Later Shchekin and Rusanov (2008) and through the years a number of collaborators more explicitly examine such systems but stop short of formulating a complete description of the intermolecular origin of what they ascribe to be a, "stabilizing disjoining pressure" (Shchekin et al., 2008, 2013; Hellmuth and Shchekin, 2015). Bruzewicz et al. (2011) observe NaCl nanoparticles but do not incorporate the particle geometry into their

theoretical model, while again relying on a phenomenological model of the short range component of the interactions. Previously, we had attempted to clearly credit and outline the previous work in the final two paragraphs of § 1 but now also return to our models advantageous in the final discussion. There we have added the text:

The formulation of continuous dry particle dissolution and droplet growth as represented by Eq. 10 and presented in the figures has several advantages over previous treatments of such systems. First we treat the intermolecular interactions explicitly in order to minimize the use of bulk parameters to model the interfacial system. The interfacial free energy minimization is then carried out and incorporated into Köhler Theory as a simple additional term that continues to allow for analytical solutions. The approach is in contrast to other treatments of analogous systems that utilize ascribed or phenomenological descriptions of short range interactions (Shchekin and Rusanov, 2008; Shchekin et al., 2008; Bruzewicz et al., 2011; Shchekin et al., 2013; Hellmuth and Shchekin, 2015) and/or do not account for the full particle geometry (Bruzewicz et al., 2011) and thus the atmospheric context.

*The inflection in some of the curves in Figure 4 warrants more explicit discussion because this behavior is not physically realistic and represents a limitation of the model. For example, in Fig. 4A, for the purple line, there are two GF values that correspond to 80% RH (at GF? 1.05 and GF? 1.3, the inflection point being ?85% RH). I was ultimately able to piece together where the inflection arises, but I encourage the authors to more explicitly discuss the trends shown in Figure 4 and their origins. Specifically, for example, I was looking for a sentence or paragraph in the discussion that explicitly stated "The inflection seen in Fig. 4 for small particles is due to..."*

• The inflection results for small particles when the strong Kelvin term effectively competes with the intermolecular interactions. However, one must remember the plotted curves represent equilibrium, and thus are not a direct prediction of what will be experimentally observed. In practice if the system humidity is gradually increased, when the inflection humidity is reached the particle is immediately exposed to a highly supersaturated environment relative to its equilibrium vapor pressure and thus will quickly grow until it reaches a new completely solvated equilibrium. In fact for small particles this very likely explains the fact that deliquescence appears spontaneous. Previously this was briefly addressed in the description of Fig. 4 but now we return to the point in the discussion. We add the text:

The model has the additional benefit of highlighting why, in practice, deliquescence is often observed to be an abrupt transition. The competition between the Kelvin term and the intermolecular forces results in an activation barrier (seen as the inflection points in Fig. 4 & Fig. 5), which when exceeded leaves a solvating particle in a highly supersaturated environment. As a result the particle grows suddenly until it reaches the new completely solvated equilibrium.

*As is well-known, atmospheric aerosols are not single component and are mixed with organic compounds. The authors mention that "natural systems may require more complex treatments. . ." (p. 6, line 5) and also that the refined Kohler theory "remains general such that its application to more complex systems may yield deeper understandings of aerosol phase state and particle behavior" (p. 7, line 12), but there is not discussion of steps that may need to be taken to apply the refined Kohler theory to mixed organic/inorganic systems. I encourage more discussion here, as it would facilitate the application of their refinement.*

• We have added the following text to the discussion:

Applying the model to more complex systems will also yield hurdles and likely make further approximations necessary, but as previously stated, may also lead to deeper insight. This model may allow some assessment of the relative importance of the short versus long range interactions and which quantities limit surface phase behavior. However, for mixtures and other materials each term of Eq. 10 would need to be re-evaluated. If bulk parameters that feed into the Köhler behavior (e.g., surface energy, water activity) are poorly constrained, strict physical interpretations will remain challenging.

*Have the authors attempted to apply their refined Kohler theory to a system other than NaCl?*

• In an ongoing collaboration with experimentalists the authors are studying a less atmospherically relevant system of butanol layers on nano-crystalline salts. However, implementing the theory for soluble particles beyond 1:1 electrolytes has not yet been treated. In that case each term in the manuscript's Eq. 10 would need to be re-evaluated, and as we state in the conclusion of §2, for more complex systems achieving analytical solutions would become challenging.

*Much of the discussion and implications of the research are mentioned for the first time in the "Conclusions" section. This discussion would be better served in its own section. For example, the discussion surrounding Figure 6, which is currently called out for the first time in the conclusions, would be appropriate in a new section for "Discussion" or "Implications". This would enhance readability.*

• The former "Conclusions" section has been split into Discussion and Conclusions sections.

*The y-axis values in the inset to Figure 3 have commas rather than periods for the decimal point.*

• The figure has been corrected.

*I believe that on p. 1, line 14, the reference Davis et al. 2015b should be labeled as "a" (and Davis et al. 2015a should then be labeled "b").*

• The reference has been corrected.

*On p. 4, line 5, Van't Hoff and E-AIM models are mentioned without references. References should be provided.*

• References have been added, such that the text now reads: "where a theoretical model (e.g., Van't Hoff – Zumdahl, 2005; E-AIM – Clegg et al., 1998; Friese and Ebel, 2010) or empirical parameterization can...".

The authors thank the reviewer for spotting the technical inconsistencies.

**References**

Biskos, G., Russell, L. M., Buseck, P. R., and Martin, S. T.: Nanosize effect on the hygroscopic growth factor of aerosol particles, Geophysical Research Letters, 33, doi:10.1029/2005GL025199, l07801, 2006.

Bruzewicz, D. A., Checco, A., Ocko, B. M., Lewis, E. R., McGraw, R. L., and Schwartz, S. E.: Reversible uptake of water on NaCl nanoparticles at relative humidity below deliquescence point observed by noncontact environmental atomic force microscopy, The Journal of Chemical Physics, 134, doi:10.1063/1.3524195, 2011.

Clegg, S. L., Brimblecombe, P., and Wexler, A. S.: Thermodynamic Model of the System $H^+$–$NH4^+$–$SO_4^{2-}$–$NO_3^-$–$H_2O$ at Tropospheric Temperatures, The Journal of Physical Chemistry A, 102, 2137–2154, doi:10.1021/jp973042r, 1998.

Ewing, G. E.: Intermolecular Forces and Clusters II, chap. $H_2O$ on NaCl: From Single Molecule, to Clusters, to Monolayer, to Thin Film, to Deliquescence, pp. 1–25, Springer Berlin Heidelberg, Berlin, Heidelberg, doi:10.1007/430_012, 2005.

Friese, E. and Ebel, A.: Temperature Dependent Thermodynamic Model of the System $H^+$–$NH_4^+$–$Na^+$–$SO_4^{2-}$–$NO_3^-$–$Cl^-$–$H_2O$, The Journal of Physical Chemistry A, 114, 11595–11631, doi:10.1021/jp101041j, 2010.

Hämeri, K., Laaksonen, A., Väkevä, M., and Suni, T.: Hygroscopic growth of ultrafine sodium chloride particles, Journal of Geophysical Research: Atmospheres, 106, 20749–20757, doi:10.1029/2000JD000200, 2001.

Hellmuth, O. and Shchekin, A. K.: Determination of interfacial parameters of a soluble particle in a nonideal solution from measured deliquescence and efflorescence humidities, Atmospheric Chemistry and Physics, 15, 3851–3871, doi:10.5194/acp-15-3851-2015, 2015.

Peters, S. J. and Ewing, G. E.: Thin Film Water on NaCl(100) under Ambient Conditions: An Infrared Study, Langmuir, 13, 6345–6348, doi:10.1021/la970629o, 1997.

Russell, L. M. and Ming, Y.: Deliquescence of small particles, The Journal of Chemical Physics, 116, 311–321, doi:10.1063/1.1420727, 2002.

Shchekin, A. K. and Rusanov, A. I.: Generalization of the Gibbs–Kelvin–Köhler and Ostwald–Freundlich equations for a liquid film on a soluble nanoparticle, The Journal of Chemical Physics, 129, 154116, doi:10.1063/1.2996590, 2008.

Shchekin, A. K., Shabaev, I. V., and Rusanov, A. I.: Thermodynamics of droplet formation around a soluble condensation nucleus in the atmosphere of a solvent vapor, The Journal of Chemical Physics, 129, 214111, doi:10.1063/1.3021078, 2008.

Shchekin, A. K., Shabaev, I. V., and Hellmuth, O.: Thermodynamic and kinetic theory of nucleation, deliquescence and efflorescence transitions in the ensemble of droplets on soluble particles, The Journal of Chemical Physics, 138, 054704, doi:10.1063/1.4789309, 2013.

Zumdahl, S. S.: Chemical Principles, Houghton Mifflin Company, Boston, MA, 5th edn., 2005.

**Review Response Supplement**

**Discussion of Fitting Parameters for NaCl**

In comment (3) Referee #1 wondered how the number of free fitting parameters could be reduced and how much more general one can make the theory by having a universal parameter set. Here we expand that discussion beyond what is relevant for the manuscript, but add it to the public discussion such that those seeking to apply the formulation within their own context might benefit.

A single parameter set can certainly be used for all sizes – and this is encouraged if one has well constrained, independent values for any of the parameters. Generally, surface charge and Hamaker constant should not depend strongly on the size of the initial particle, something which we also observe from the weak variations in the fitted values as presented in the main text in Table 1. However, the saturated equilibrium concentration $C$ may have a size dependency. For example, using the three parameter fitting the brine concentration for the Biskos et al. (2006) data decreases as shown in Figure RS1. From such fitted values an empirical relationship for $C$ can be derived, which in this case it takes the form of a decaying exponential (Figure RS1).

[Figure]

Figure RS1: Fitted concentration of the brine layer as a function of particle size for the Biskos et al. (2006) data, as taken from Table 1 in the main text.

In Figure RS2 a number of alternative equilibrium solutions are plotted based on different fitting procedures for a subset of the Biskos et al. (2006) data. What these plots make clear is that although, as one would expect, the best fits occur with the maximum three free parameters, fitting with up to two of the parameters fixed still yields high quality data representations. However, the quality of the data prediction is also clearly linked to how well quantified the parameters are. For the system we have presented, the breadth of independent information regarding surface charge, Hamaker constant, and equilibrium brine concentration, especially for the smallest particles is limited and thus using fixed literature values for $q_s$ and $A_h$, while fitting the $C$ value leads to poor data representations (as in panel (d)). Using fixed values for all three parameters yields even poorer results (not shown) which are clearly unphysical.

[Figure]

[Figure]

(a) Biskos et al. (2006) data reploted with the three-free parameter non-linear least squares, equilibrium solutions plotted in matching colors. Identical to Figure 4 in the main text.

(b) Theoretical equilibrium solutions reploted using fixed $q_s$ and $A_h$, taken to be the average values as calculated from the Table 1 data in the main text, and a variable $C$ as predicted from the presented empirical equation (see also Figure RS3).

[Figure]

[Figure]

(c) Theoretical equilibrium solutions again replotted. Here $q_s$ and $A_h$ are again taken to be the average values as calculated from the Table 1 data in the main text. $C$ is treated as a free parameter and fit using a least squares minimization (see also Figure RS3).

(d) Alternative solution to Figure RS2c where $q_s$ and $A_h$ are fixed and assigned the reference values determined from the literature. $C$ is again treated as a free parameter and fit using a least squares minimization (see also Figure RS3).

Figure RS2: Reploting Biskos et al. (2006) data as it appeared in the main text Figure 4 using, (a) 3 free parameters in identical manner to Figure 4, (b) 0 free fitting parameters, (c) 1 free fitting parameter with average values for fixed parameters, and (d) 1 free fitting parameter with fixed values taken from literature sources.

The table presented as Figure RS3 summarizes the quantities used to generate the plots in Figure RS2.

| R_s [nm] | Figure RS2a | | | Figure RS2b | | | Figure RS2c | | | Figure RS2d | | |
|---|---|---|---|---|---|---|---|---|---|---|---|---|
| | $q_k$ [C/m$^2$] | $A_h$ [J] | C [mol/m$^3$] | $q_k$ [C/m$^2$] | $A_h$ [J] | C [mol/m$^3$] | $q_k$ [C/m$^2$] | $A_h$ [J] | C [mol/m$^3$] | $q_k$ [C/m$^2$] | $A_h$ [J] | C [mol/m$^3$] |
| Fitting Procedure | NLF | NLF | NLF | Average | Average | Empirical | Average | Average | NLF | Kobayashi 2014 | Hansen-Goos 2014 | NLF |
| 3 | 0.3 | -7.5E-20 | 8800 | | | 8873 | | | 9050 | | | 9300 |
| 4 | 0.3 | -4E-20 | 8000 | | | 8314 | | | 8000 | | | 8000 |
| 5 | 0.4 | -2E-20 | 7620 | | | 7857 | | | 7500 | | | 7800 |
| 7.5 | 0.002 | -3E-20 | 7020 | 0.1665 | -5.2E-20 | 7043 | 0.1665 | -5.2E-20 | 7020 | 0.12 | -1.5E-20 | 7020 |
| 10 | 0.01 | -1.5E-19 | 6500 | | | 6550 | | | 6500 | | | 6600 |
| 15 | 0.02 | -4E-20 | 6300 | | | 6070 | | | 6300 | | | 6300 |
| 20 | 0.3 | -5.5E-20 | 6000 | | | 5893 | | | 6000 | | | 6000 |
| 30 | 0 | -2E-21 | 5800 | | | 5804 | | | 5600 | | | 5800 |

Figure RS3: Parameters used to plot Figure RS2. The parameters were determined in three different ways. **NLF**: Parameters are determined using a non-linear least squares minimization. **Average**: Average values calculated from corresponding NLF columns. **Emprical**: Values calculated from the empirical equation determined in Figure RS1: $C = 5617.5 \exp(-2 \times 10^8 R_s) + 5790$